# Construction of a Recombinant Porcine Epidemic Diarrhea Virus Encoding Nanoluciferase for High-Throughput Screening of Natural Antiviral Products

**DOI:** 10.3390/v13091866

**Published:** 2021-09-18

**Authors:** Wan Li, Mengjia Zhang, Huijun Zheng, Peng Zhou, Zheng Liu, Anan Jongkaewwattana, Rui Luo, Qigai He

**Affiliations:** 1State Key Laboratory of Agricultural Microbiology, College of Veterinary Medicine, Huazhong Agricultural University, Wuhan 430070, China; liwan@webmail.hzau.edu.cn (W.L.); zhangmengjia@webmail.hzau.edu.cn (M.Z.); jduy@webmail.hzau.edu.cn (H.Z.); zhoupeng0603@webmail.hzau.edu.cn (P.Z.); he628@mail.hzau.edu.cn (Q.H.); 2Key Laboratory of Preventive Veterinary Medicine in Hubei Province, the Cooperative Innovation Center for Sustainable Pig Production, Wuhan 430070, China; 3Key Laboratory of Pesticide & Chemical Biology of Ministry of Education, Hubei International Scientific and Technological Cooperation Base of Pesticide and Green Synthesis, International Joint Research Center for Intelligent Biosensing Technology and Health, College of Chemistry, Central China Normal University, Wuhan 430079, China; liuz1118@mail.ccnu.edu.cn; 4Virology and Cell Technology Laboratory, National Center for Genetic Engineering and Biotechnology (BIOTEC), National Science and Technology Development Agency (NSTDA), 113 Thailand Science Park, Phahonyothin Rd., Pathumthani 12120, Thailand; Anan.jon@biotec.or.th

**Keywords:** PEDV, nanoluciferase, high-throughput screening, antiviral compounds, reverse genetics system

## Abstract

Porcine epidemic diarrhea virus (PEDV) is the predominant cause of an acute, highly contagious enteric disease in neonatal piglets. There are currently no approved drugs against PEDV infection. Here, we report the development of a nanoluciferase (NLuc)-based high-throughput screening (HTS) platform to identify novel anti-PEDV compounds. We constructed a full-length cDNA clone for a cell-adapted PEDV strain YN150. Using reverse genetics, we replaced the open reading frame 3 (ORF3) in the viral genome with an NLuc gene to engineer a recombinant PEDV expressing NLuc (rPEDV-NLuc). rPEDV-NLuc produced similar plaque morphology and showed similar growth kinetics compared with the wild-type PEDV in vitro. Remarkably, the level of luciferase activity could be stably detected in rPEDV-NLuc-infected cells and exhibited a strong positive correlation with the viral titers. Given that NLuc expression represents a direct readout of PEDV replication, anti-PEDV compounds could be easily identified by quantifying the NLuc activity. Using this platform, we screened for the anti-PEDV compounds from a library of 803 natural products and identified 25 compounds that could significantly inhibit PEDV replication. Interestingly, 7 of the 25 identified compounds were natural antioxidants, including Betulonic acid, Ursonic acid, esculetin, lithocholic acid, nordihydroguaiaretic acid, caffeic acid phenethyl ester, and grape seed extract. As expected, all of the antioxidants could potently reduce PEDV-induced oxygen species production, which, in turn, inhibit PEDV replication in a dose-dependent manner. Collectively, our findings provide a powerful platform for the rapid screening of promising therapeutic compounds against PEDV infection.

## 1. Introduction

Porcine epidemic diarrhea virus (PEDV), a member of swine enteric coronaviruses (CoVs), causes diarrhea, vomiting, anorexia, and dehydration in neonatal piglets [1]. After PEDV classical strains of the genogroup I (GI) were first reported in the UK in 1971 [2], it has spread throughout most European and Asian countries [3,4]. In 2010, highly pathogenic PEDV variants suddenly emerged in China, and they rapidly spread to North America, Europe, and other Asian countries in 2013 [5,6,7]. Since then, virulent PEDV variants (GII) have caused epidemic outbreaks, resulting in high morbidity and mortality rates in newborn piglets [8]. Currently, PEDV is considered one of the most devastating swine viruses, resulting in massive economic losses to the global pig industry [6,8,9]. To prevent and control the outbreaks, several commercial vaccines, such as inactivated, live-attenuated, and S subunit vaccines, have been approved for use against PEDV in China, South Korea, and the USA [10,11,12]. Nevertheless, these vaccines cannot provide adequate protective immunity against the prevalent PEDV strains, and large-scale outbreaks of PEDV still occur and frequently recur [13]. Thus, the development of antiviral agents is urgently required to prevent PEDV infection from compensating for the vaccine’s effect.

Natural products extracted from plants, fungi, and bacteria are increasingly pursued as potential alternative antiviral agents. A limited number of natural compounds, such as quercetin [14], quercetin 7-rhamnoside [15], glycyrrhizin [16], surfactin [17], and aloe extract [18], have been demonstrated to exhibit antiviral activities against PEDV infection. These studies have successfully identified natural compounds against PEDV mainly by quantifying virus titers in compound-treated cells by traditional infection- or RT-PCR-based assays. However, such methods are laborious, time-consuming, and challenging to achieve high-throughput screening (HTS) of compounds against PEDV. Incorporating a reporter gene into the viral genome can rapidly and effectively monitor viral replication, which provides an ideal tool for HTS and evaluating antiviral compounds. To date, several reporter CoVs have been genetically engineered for the high-throughput screening of anti-CoV compounds and agents. For instance, Shen and colleagues recently screened a 2000-compound library of drugs using the recombinant human CoV OC43 (HCoV-OC43) expressing Renilla luciferase and identified 36 inhibitors against HCoV-OC43 replication in vitro [19].

PEDV is an enveloped, single-stranded positive-sense RNA virus belonging to the genus α-coronavirus of the family Coronaviridae [20]. The PEDV genome is about 28 kb in length, which contains a 5′ untranslated region (UTR), seven open reading frames (ORFs) with the order 5′-ORF1a/1b-S-ORF3-E-M-N-3′, and a 3′ UTR. Two large overlapping ORFs, ORF1a, and ORF1b occupy approximately two-thirds of the genome encoding polyproteins pp1a and pp1b that are further cleaved by viral proteases into 16 nonstructural proteins involved in viral genome replication and transcription [21]. The remaining one-third of the genome encodes four structural proteins responsible for virion assembly, including spike (S), envelope (E), membrane (M), and nucleocapsid (N) proteins, and one accessory ORF3 protein [21,22]. Notably, ORF3, positioned between the S and E genes, is the only identified accessory gene in PEDV. Although the ORF3 protein has been demonstrated to bind directly to the S protein and might be involved in virus assembly [23], the ORF3 protein is dispensable for virus replication in vitro. Using targeted RNA recombination, Li et al. replaced the ORF3 gene of the attenuated PEDV DR13 (GI) with the GFP and Renilla luciferase genes and applied these report PEDVs to virus neutralization assay [24].

NanoLuc luciferase (NLuc) is a small, highly stable, and ATP-independent enzyme that has higher specific activity and produces a more vigorous bioluminescence signal intensity (>150-fold increase) in comparison to the Firefly and Renilla luciferases [25]. In this study, we established a reverse genetics system of the PEDV YN150 strain, a prevalent GII strain, and replaced the entire ORF3 gene with the NLuc for engineering a recombinant PEDV expressing NLuc (rPEDV-NLuc). We found that the growth kinetics of rPEDV-NLuc was comparable to that of wild-type PEDV, and the level of luciferase activity could be stably detected in rPEDV-NLuc-infected cells. Remarkably, the level of luciferase activity exhibited a strong positive correlation with viral titers in rPEDV-NLuc-infected cells, providing a powerful tool for monitoring the extent of viral replication. Finally, we employed the rPEDV-NLuc to perform a high-throughput screening (HTS) of the anti-PEDV compounds from a library of 803 natural products and successfully identified 25 compounds that could potently inhibit PEDV replication.

## 2. Materials and Methods

### 2.1. Cells, Viruses, and Antibodies

Vero cell (#CCL-81) was purchased from American Type Culture Collection (ATCC, Manassas, VA, USA) and cultured in Dulbecco’s modified Eagle’s medium (DMEM, Gibco, Thermo Fisher Scientific) supplemented with 10% fetal bovine serum (FBS). The cell-culture-adapted PEDV YN150 strain (GenBank accession number MZ581326) was obtained by serial passages of the virulent PEDV YN1 strain (GenBank accession number KT021227). PEDV YN150 was propagated in Vero cells in DMEM, containing 10% tryptose phosphate broth (TPB) and 10 μg/mL trypsin. The anti-PEDV S mAb 4B2, anti-PEDV N mAb 8E2, and anti-NLuc mAb 5H6 used in this study were generated in our laboratory.

### 2.2. Construction of a Full-Length cDNA Clone

The full-length cDNA clone of the PEDV YN150 strain was constructed using a unidirectional molecular clone strategy, which was previously described for the PC22A strain of PEDV [26]. Briefly, the full-length PEDV cDNA clone was divided into six contiguous fragments that could be systematically linked by five BsmBI restriction sites at nucleotide positions 6055, 9880, 14880, 19216, and 23,497. A T7 promoter sequence and 22 A residues were added to the viral genome’s immediate 5′ end and 3′ end. Five naturally occurring BsmBI sites at nucleotide positions 376, 787, 2842, 11564/11566, and 23,882 were removed by introducing synonymous mutations to ensure the appropriate assembly of the full-length infectious clone. Six cDNA fragments (F1–6) were synthesized and cloned into the pJET1.2/blunt cloning vector (Thermo Scientific, Waltham, MA, USA). After propagation in *Escherichia coli*, all PEDV fragments were sequenced to ensure their sequence fidelity. These fragments were digested using restriction enzymes, separated through 0.8% agarose gels, excised, and purified using a QIAquick gel extraction kit (Qiagen, Hilden, Germany). The digested fragments were ligated overnight at 4 °C using T4 DNA ligase (Thermo Scientific), extracted with phenol/chloroform, and precipitated with isopropanol. Full-length PEDV RNA was generated by in vitro transcription using mMESSAGE mMACHINE T7 Transcription Kit (Ambion, Austin, TX, USA) according to the manufacturer’s instruction. SP6 PEDV N gene transcripts were generated from the PCR-purified PEDV N gene sample using a 4:1 ratio of cap to GTP (Ambion).

To construct the PEDV-NLuc with the ORF3 deletion, ORF3 in the PEDV-F6 fragment was replaced with synthesized NLuc gene using fusion PCR. PEDV-F6-△ORF3-NLuc was amplified in *Escherichia coli* and sequenced to guarantee seamless substitution of ORF3 with NLuc and the retainment of the transcription regulatory sequence (TRS) of ORF3.

### 2.3. In Vitro Transfection

The 30 µL of the full-length PEDV RNA and 10 µL of N RNA transcripts were together transfected into Vero cells (8.0 × 10^6^ cells/mL) in phosphate-buffered saline (PBS), and three electrical pulses of 450 V at 50 µF were given by a Gene Pulser X cell electroporation system (Bio-Rad, Hercules, CA, USA). The transfected cells were seeded into a 75 cm^2^ flask and incubated at 37 °C. After 12 h, the cells were washed with PBS and incubated in DMEM containing 10 μg/mL trypsin. Rescued viruses were harvested and further purified by plaque assay.

### 2.4. Virus Genome Sequencing

Viral RNAs were extracted from the supernatant of PEDV-infected Vero cells using TRIzol Reagent (Invitrogen, Waltham, MA, USA). Viral genome RNA was converted to cDNA by Transcriptor First Strand cDNA Synthesis Kit (Roche, Basel, Switzerland). The full-length genome of rescued PEDVs was sequenced by conventional PCR with 28 pairs of primers, listed in Appendix A.

### 2.5. Indirect Immunofluorescence Assay (IFA)

PEDV-infected Vero cells were fixed with 4% paraformaldehyde for 15 min and subsequently permeabilized with 0.1% Triton X-100 (Sigma, St. Louis, MO, USA) for 10 min. After blocking with 5% bovine serum albumin (BSA) in PBS, the cells were incubated with anti-PEDV S mAb 4B2, followed by AlexaFluor 488 donkey anti-mouse IgG (Invitrogen). Cells were stained with 4′,6-diamidino-2-phenylindole (Invitrogen) for 15 min, and the fluorescence signal was examined using a fluorescence microscope.

### 2.6. Luciferase Assay

Vero cells seeded in 96-well plates were infected with rPEDV-NLuc at the indicated viral amount. At 20 h post-infection, infected cells were lysed by Passive Lysis Buffer (Promega, Madison, WI, USA) and assayed for luciferase activity using the Nano-Glo Luciferase Assay System (Promega) according to the manufacturer’s instructions.

### 2.7. Western Blotting

PEDV- and rPEDV-NLuc-infected cell lysates were obtained using lysis buffer (65 mM Tris-HCl (pH 6.8), 4% sodium dodecyl sulfate, 3% DL-dithiothreitol, and 40% glycerol), separated by 12% SDS-PAGE, transferred to polyvinylidene difluoride membranes, and blocked with 10% skim milk. The membranes were incubated with anti-PEDV N mAb 8E2 and anti-Nluc mAb 5H6 followed by horseradish peroxidase (HRP)-conjugated goat anti-mouse IgG (ABclonal, Wuhan, China). Signals were detected using the SuperSignal West Pico Luminal kit (Pierce, Woodland Hills, CA, USA).

### 2.8. HTS of a Natural Product Library

A library of 803 natural products was purchased from Selleck Chemicals (Houston, TX, USA). Compounds were stored in 10 mM stock solution in DMSO at −80 °C until use. The primary HTS assay was performed in Vero cells. Briefly, Vero cells were seeded into 96-well plates in DMEM containing 10% FBS. After 12 h, 1 μL of each compound was added to the cell cultures in plates at a final concentration of 10 μM (one compound per well) in triplicate; 1 μL of DMSO was used for the controls. After 1 h incubation, the cells were washed three times and inoculated with 0.01 MOI rPEDV-NLuc diluted in 100 μL/well maintenance medium containing 10 μM compound and 10 μg/mL trypsin. After 20 h incubation, the luciferase activity in infected cells was determined using the Nano-Glo Luciferase Assay System (Promega). The primary candidates were identified using the criteria of no apparent cytotoxicity and an average >90% inhibition in triplicate wells. The criteria of dose-dependent inhibition and cell viability of >80% were used to reconfirm the candidates. Moreover, the compound-specific toxicity (CC_50_) was calculated using GraphPad Prism 6, and compounds with an SI > 10 were considered hits in this study. Cytotoxicity was assessed by the 3-(4,5-dimethyl-2-thiazolyl)-2,5-diphenyl-2H-tetrazolium bromide (MTT) assay.

### 2.9. Detection of ROS Production

The intracellular ROS production was examined using the fluorescent probe 6-carboxy-2′, 7′-dichlorodihydrofluorescein diacetate (DCFH-DA, Beyotime Biotechnology, Shanghai, China). Following PEDV infection, Vero cells were washed with PBS and then treated with 10 μM DCFH-DA for 20 min at 37 °C. After treatment, cells were washed twice in PBS and subjected to fluorescence microscopy observation and FACS analysis.

## 3. Results

### 3.1. Construction of the Full-Length cDNA Clone of PEDV

Recently, the reverse genetics system for the PEDV PC22A strain has been successfully developed based on the unidirectional assembly of a full-length genome cDNA from a set of consecutive shorter cDNA fragments [26]. Here, we employed a similar strategy to construct the full-length cDNA clone of PEDV YN150. The complete viral genome was divided into six contiguous fragments (F1–6) that could be systematically joined by five BsmBI restriction sites at nucleotide positions 6055, 9880, 14,880, 19,216, and 23,497 (Figure 1A). Five naturally occurring BsmBI sites at nucleotide positions 379, 790, 2834, 11,557, and 23,886 located at F1, F3, and F6 were removed by silent mutations but not changing protein products. Notably, BsmBI is a type-IIS restriction endonuclease that cleaves outside their recognition site and leaves unique 4-nucleotide overhangs, allowing for the systematic, efficient, and directional assembly of the six smaller cDNA fragments into the full-length PEDV genomic cDNA by in vitro ligation. A T7 promoter sequence and 22 A residues were added to the immediate 5′ end and 3′ end of the viral genome (Figure 1A) for in vitro transcription and polyadenylation of RNA transcripts.

### 3.2. Recovery, Identification, and Characterization of rPEDV

Each PEDV fragment was digested, purified, and ligated to generate a full-length viral cDNA genome. PEDV transcripts were then synthesized with the T7 RNA polymerase using the full-length cDNA as a template. Since the supplement of N gene transcript can facilitate the replication efficiency of several CoVs, such as MHV, TGEV, SARS-CoV, and MERS-CoV [27,28,29,30], capped PEDV-N gene transcripts were mixed with the full-length transcripts, and co-electroporated into Vero cells. Apparent cytopathic effects (CPE) were observed within 30 h post-transfection, and the expression of the S protein could be detected by IFA using anti-PEDV S mAb (Figure 1B). After plaque purification, the complete genome of rPEDV was sequenced. We observed that the whole genome sequence of rPEDV was identical to the cDNA clone with no unwanted mutation, including the five genetic markers, as shown in Figure 1C. These results indicated the successful rescue of rPEDV in Vero cells.

### 3.3. Construction and Rescue of rPEDV-NLuc

To express NLuc from the PEDV genome, the ORF3 gene in the F6 fragment was replaced with that of NLuc (Figure 2A). Transcription regulatory sequences (TRS) of ORF3 and E were preserved for initiating subgenomic RNA expression (Figure 2A). After in vitro ligation and transcription, the full-length transcripts were electroporated into Vero cells for rPEDV-NLuc recovery. Similar to the wild-type PEDV, detectable CPE caused by rPEDV-NLuc was readily observed within 30 h post-transfection (Figure 2B). The expression of S and N proteins was also detected in Vero cells infected with the wild-type PEDV and rPEDV-NLuc by IFA and Western blotting (Figure 2B,C). After plaque purification, we sequenced the whole genome of rPEDV-NLuc and confirmed that the ORF3 was successfully replaced by NLuc in rPEDV-NLuc without unwanted nucleotide change compared to its cDNA clone (Figure 2D). Notably, we observed that rPEDV-NLuc retained the inserted NLuc gene up to five passages in Vero cells (data not shown). To investigate whether the presence of NLuc affects viral replication, we determined the viral plaque size and growth kinetics of rPEDV-NLuc. As shown in Figure 2E, rPEDV-NLuc produced similar plaque morphology with the wild-type PEDV and rPEDV. Titers of rPEDV-NLuc at the different time points post-infection were also similar to those of the wild-type PEDV and rPEDV (Figure 2F). These results suggested that the replacement of ORF3 with exogenous NLuc in PEDV YN150 did not affect viral proliferation in Vero cells.

### 3.4. Assessment of rPEDV-NLuc Replication via Luciferase Assay

To further confirm the expression of NLuc, we examined the bioluminescent signal in cells infected with rPEDV-NLuc. As shown in Figure 3A, plaques produced by rPEDV-NLuc could be easily visualized as luminescent signals using the In Vivo Imaging System (IVIS). Bioluminescence signals of Vero cells infected with rPEDV-NLuc were also enhanced with the increased MOI (Figure 3B). To investigate whether the level of NLuc activity could accurately reflect the extent of viral replication, we detected the NLuc activity in cell lysates infected with rPEDV-NLuc at different times post-infection. The NLuc activity could be measured as early as 4 h post-infection and increased until 28 h post-infection (Figure 3C), which has similar dynamics of rPEDV-NLuc growth determined by TCID_50_ as presented in Figure 2F. Additionally, an increased amount of rPEDV-NLuc infection could result in elevated NLuc activity in a dose-dependent manner. A linear log RLU-log PFU correlation was observed in the cells infected with rPEDV-NLuc (Figure 3D). These data indicated that the level of viral replication could be effectively monitored by measuring the NLuc activity in the Vero cells infected with rPEDV-NLuc.

### 3.5. Screening of a Natural Product Library for Inhibitors of PEDV Infection

Using the rPEDV-NLuc reporter virus, we optimized the HTS assay conditions to be 10,000 Vero cells/well in 96-well plates and rPEDV-NLuc infection at an MOI of 0.01. Under this condition, the coefficient of variation (CV) and Z factor were 2.5% and 0.94, respectively, demonstrating that the assay was robust and suitable for HTS.

A flowchart of the HTS is depicted in Figure 4A. In the primary screening from the 803 compounds library at a concentration of 10 μM, 63 hits were found to significantly inhibit rPEDV-NLuc replication with 90% reduced NLuc activity and no apparent cytotoxicity (Figure 4B). Among these prime candidates, 25 compounds were selected based on their dose-dependent inhibition and a selective index (SI) > 10 (Appendix A, Table 1). To verify the results obtained by the luciferase reporter assays, we confirmed the antiviral effect of the 25 hits in Vero cells using wild-type PEDV. Consistent with the HTS results, all 25 compounds significantly reduced PEDV replication with at least a 1-log-unit decrease in the viral titers at the concentration of 10 μM (Figure 4C).

Of note, we observed that 7 of the 25 identified compounds had been reported to possess antioxidant activity, including Betulonic acid (BA) [31], Ursonic acid (UA) [32], esculetin [33], lithocholic acid (LCA) [34], nordihydroguaiaretic acid (NDGA) [35], caffeic acid phenethyl ester (CAPE) [36] and grape seed extract (GSE) [37]. To further verify the antiviral activities of the natural antioxidants, we examined whether the natural antioxidants could inhibit PEDV replication in a dose-dependent manner via TCID50 and IFA. Compared with the DMSO group, all seven antioxidants significantly inhibited PEDV proliferation at 2.5, 5, and 10 μM concentrations, and the dose-dependent inhibition of viral replication were also observed in the cells treated with all antioxidants (Figure 5A,B). Notably, all seven antioxidants at 2.5 μM are efficient in reducing PEDV replication with at least a 60% decline in viral load, suggesting that these antioxidants exhibited the potent inhibition of PEDV proliferation.

### 3.6. Seven Natural Products Inhibit PEDV-Induced Reaction Oxygen Species (ROS)

Recent studies showed that PEDV infection induces ROS accumulation in Vero cells [38]. We further explored whether the seven natural antioxidants can suppress the PEDV-induced ROS production. Using fluorescence microscopy, we observed that PEDV infection dramatically increased DCFH-DA fluorescence as expected. In contrast, only a few fluorescence signals were detected in PEDV-infected cells when treated with BA, UA, esculetin, LCA, NDGA, CA, and GSE, respectively (Figure 6A). To further investigate the effect of the natural antioxidants on PEDV-induced ROS, we employed flow cytometry to detect the ROS levels in PEDV-infected cells after the treatment of antioxidants. As shown in Figure 6B, ROS production induced by PEDV was significantly decreased in the cells treated with these natural antioxidants.

## 4. Discussion

PEDV has been considered one of the most devastating porcine viruses that have emerged or re-emerged in recent decades, causing severe economic losses in the global pork industry. Even though both inactivated and live-attenuated vaccines have been widely used to control the prevalence of PEDV in China and other counties, the PEDV outbreak is still ongoing, and novel PEDV strains were continually detected in vaccinated pig herds [3,8]. The development of effective antiviral agents is another strategy to control PEDV infection. Salinomycin [39] and remdesivir [40] have been reported as potential antiviral drugs against PEDV infection in vitro. Several natural products extracted from plants, such as glycyrrhizin [16], quercetin 7-rhamnoside [15], and epigallocatechin-3-gallate [41], have been demonstrated to reduce PEDV infection in Vero cells. To search for new anti-PEDV agents, Deejai et al. screened the FDA-approved drugs that could associate with the PEDV N protein using virtual screening and found that trichlormethiazide, D-(+) biotin and acetazolamide could inhibit PEDV replication in vitro [42]. Recently, based on the assessment of CPE and viral protein expression by IFA, Wang et al. screened a natural product library and demonstrated that tomatidine could reduce PEDV proliferation in Vero cells [4].

NLuc is a novel engineered luciferase that is smaller and brighter than firefly or Renilla luciferase (about a 150-fold increase in luminescence). It utilizes the synthetic substrate furimazine to produce high-intensity, glow-type luminescence but low background activity [25]. Recently, Xie et al., constructed a recombinant SARS-CoV-2 expressing NLuc that can rapidly detect neutralization antibodies and screen antiviral drugs [43]. In this study, we employed the reverse genetics system to engineer a recombinant PEDV expressing the NLuc gene (Figure 2). PEDV ORF3 is the only accessory ORF that encodes a putative ion channel protein [44]; however, this is dispensable for viral replication in vitro and in vivo [24]. During serial passaging, PEDV ORF3 usually becomes abortive due to deletion mutations [45,46,47]. In the cell-culture adapted YN150 strain, a nucleotide deletion within the ORF3 gene leads to early termination of ORF3 at 144 aa. Therefore, the ORF3 of PEDV YN150 is suitable for replacement with the NLuc gene. As expected, we found that the NLuc gene was stably incorporated into the genome of PEDV YN150 with no apparent effects on viral kinetics or plaque phenotype (Figure 2). Bioluminescence signal produced during rPEDV-NLuc infection could accurately reflect the extent of viral replication (Figure 3). The luciferase levels during rPEDV-NLuc infection showed an excellent linear relationship with the viral titers (Figure 3), indicating that bioluminescence signal produced during rPEDV-NLuc infection can be used to monitor viral replication in Vero cells.

Using the engineered rPEDV-NLuc, we performed HTS of an 803-compound library of natural products and identified 25 compounds exhibiting anti-PEDV activity (Figure 4). Intriguingly, we discovered that 7 of the 25 identified inhibitors are antioxidants, including Betulonic acid, Ursonic acid, esculetin, lithocholic acid, nordihydroguaiaretic acid, caffeic acid, and grape seed extract (Figure 4). All seven antioxidants were confirmed to inhibit wild-type PEDV replication in a dose-dependent manner in Vero cells (Figure 5). Even at 2.5 μM concentration, all selected antioxidants showed a level of inhibition of at least 60% (Figure 5). Previous studies demonstrated that infection of Vero cells with PEDV causes ROS accumulation in a time-dependent manner [38]. Here, we showed that the seven identified antioxidants could dramatically reduce PEDV-trigger ROS accumulation, and their inhibitory effect of ROS accumulation was more substantial than that of a well-known ROS inhibitor N-acetyl-L-cysteine (NAC) (Figure 6). NAC was found to limit the replication of influenza virus (IV) and reduce IV-induced apoptosis [48,49,50]. In our study, NAC also exhibited anti-PEDV activity in a dose-dependent manner (Figure 5), suggesting that ROS production was involved in PEDV replication. More recently, Sun et al. reported that ROS accumulation played a critical role in ER stress and autophagy during PEDV infection [51]. In our previous study, PEDV-mediated autophagy facilitated viral replication [52]. We speculated that the seven hit antioxidants reduced PEDV-induced ROS accumulation, probably leading to the cells being defective in autophagy, thus impairing viral replication. Nevertheless, the precise mechanism by which antioxidants inhibit PEDV replication remains to be determined.

Besides the antioxidants, two hit compounds, protoporphyrin IX and 7-Ethylcamptothecin, exhibited inhibitory activities on the replication of PEDV with an SI of >80.7-Ethylcamptothecin was reported as an anti-cancer chemical with potent activity against various murine tumors via topoisomerase inhibition [53]. To our knowledge, there has been no report on the antiviral activity of 7-Ethylcamptothecin; however, it showed anti-PEDV activity with IC_50_ of <1.25 μM. Notably, protoporphyrin IX can insert into the lipid vesicles of the vesicular stomatitis virus and impair the viral membrane organization, leading to the inhibition of viral replication [54]. Recently, Gu et al., reported that protoporphyrin IX could completely inhibit the cytopathic effect caused by SARS-CoV-2. They speculated protoporphyrin IX probably binds with the receptor ACE2 and blocks the S protein-mediated entry [55,56]. Whether protoporphyrin IX inhibits PEDV entry needs further exploration.

In conclusion, our study has established a powerful reverse genetics platform for the PEDV YN150 strain and engineered a reporter PEDV expressing the NLuc (rPEDV-NLuc). We found that the recombinant virus showed similar viral plaque size and growth kinetics to the parental virus in vitro. The bioluminescence signal produced during rPEDV-NLuc infection could accurately reflect the level of viral replication in Vero cells. Using the engineered virus, we performed the HTS of anti-PEDV compounds from a library of 803 natural products and successfully identified 25 compounds inhibiting the replication of PEDV. Our findings provide a powerful tool for the rapid screening of antiviral drugs and neutralization antibody detection and offer new and promising therapeutic strategies against PEDV infection.

## Figures and Tables

**Figure 1 viruses-13-01866-f001:**
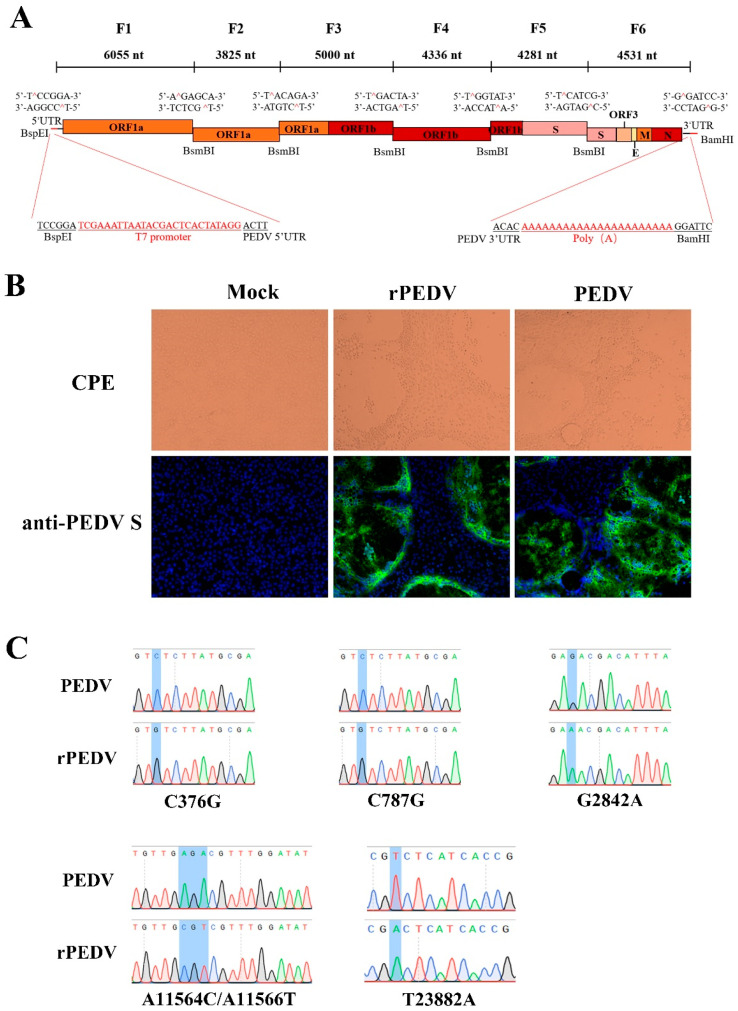
The assembly of the full-length PEDV cDNA clone and recovery of rPEDV. (**A**) The schematic diagram for the assembly of the full-length cDNA clone of PEDV YN150. The full-length genome of the PEDV YN150 was divided into six contiguous cDNAs (F1–6). Restriction sites linking fragments are noted. (**B**) Wild-type PEDV, rPEDV, or mock-infected Vero cells were subjected to IFA using anti-PEDV S mAb 4B2 at 20 hpi (magnification, ×100). (**C**) Five genetic markers were verified by sequencing rPEDV genome. Four genetic markers at nucleotide positions 376, 787, 2842, and 11,564/11,566 are located on the ORF1ab, while one genetic marker at nucleotide position 23,882 is located on the Spike.

**Figure 2 viruses-13-01866-f002:**
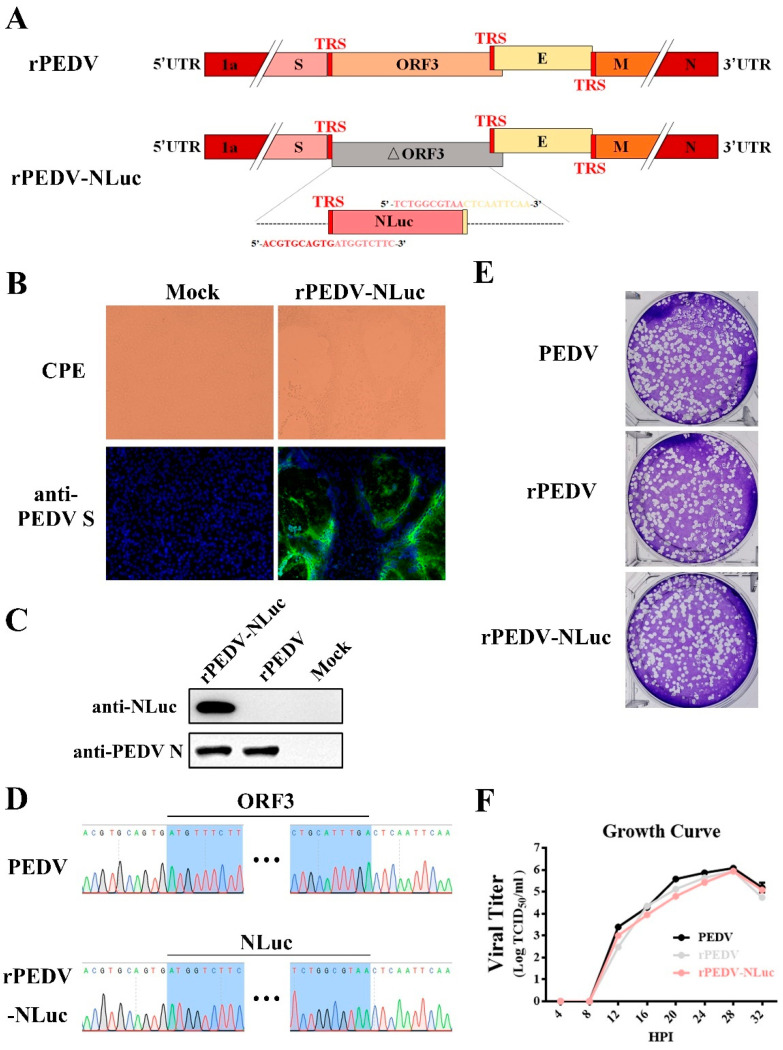
Rescue of rPEDV-NLuc. (**A**) Schematic representation of the cDNA clone of rPEDV-NLuc. (**B**) The rPEDV-NLuc- and mock-infected Vero cells were detected by IFA using anti-PEDV S mAb 4B2 (magnification, ×100). (**C**) Western blot analysis of the expression of PEDV N and NLuc using the anti-PEDV N mAb 8E2 and the anti-NLuc mAb 5H6, respectively. (**D**) Identification of the ORF3 replacement with NLuc by sequencing the rPEDV-NLuc genome. (**E**) Representative plaques of wild-type PEDV-, rPEDV-, and rPEDV-NLuc-infected Vero cell. (**F**) Comparison of growth kinetics of wild-type PEDV, rPEDV, and rPEDV-NLuc. Vero cells were infected with wild-type PEDV, rPEDV, and rPEDV-NLuc at an MOI of 0.01, and viral titers were detected at indicated time points by TCID_50_ assay. Data are represented as the mean ± SD (*n* = 3).

**Figure 3 viruses-13-01866-f003:**
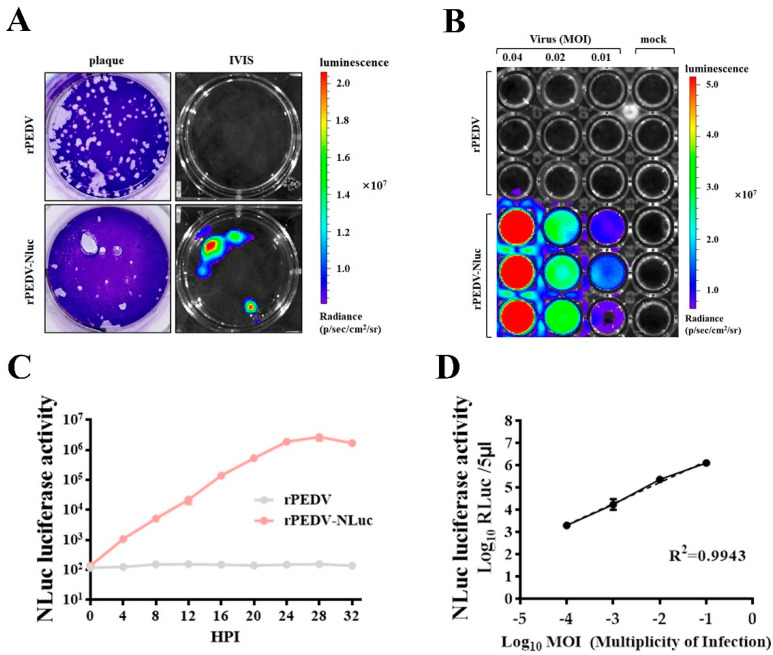
Assessment of rPEDV-NLuc replication via luciferase assay. (**A**) Nanoluciferase imaging of plaques from Vero cells infected with rPEDV-NLuc. Vero cells were infected with rPEDV-NLuc at an MOI of 0.001 or rPEDV at an MOI of 0.01. At 36 h post-infection, plaques formed by PEDVs were visualized as luminescent signals using the In Vivo Imaging System (IVIS). (**B**) Vero cells were infected with rPEDV and rPEDV-NLuc at different MOIs. At 20 h post-infection. NLuc substrate was added to each well and luminescent signal were detected by IVIS. (**C**) The replication kinetics of rPEDV-NLuc was quantified by luminometry. Vero cells were infected with rPEDV and rPEDV-NLuc at an MOI of 0.01, and the luminescent signals were quantified by luminometry at the indicated times. Data are represented as the mean ± SD (*n* = 3). (**D**) Vero cells were infected with rPEDV-NLuc at the different MOIs, and the luminescent signals of cell lysates were tested at 20 h post-infection by the Nano-Glo Luciferase Assay System (Promega). The analysis of correlation between virus input and bioluminescent signal was analyzed by GraphPad Prism 7.0. Data are represented as the mean ± SD (*n* = 3).

**Figure 4 viruses-13-01866-f004:**
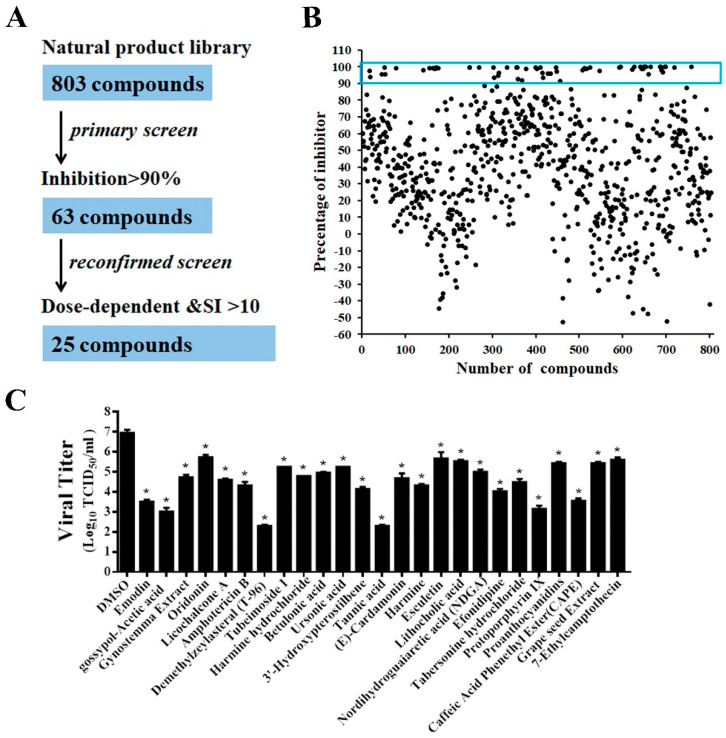
HTS for inhibitors of PEDV infection from a natural products library. (**A**) HTS assay flowchart. The 63 primary candidates were identified using criteria of an average >90% inhibition in triplicate wells. In the secondary screening, 25 compounds with dose-dependent inhibition and SI >10 was selected. (**B**) The HTS of a library of 803 natural products for primary candidates against PEDV infection. Each dot represents the percentage inhibition of a compound at a concentration of 10 μM. Dots in the blue box represents the prime candidates with inhibition >90% and no obvious cytotoxicity. (**C**) Confirmation of anti-PEDV activity by TCID_50_. Vero cells were treated with the hit compounds for 1 h and then inoculated with wild-type PEDV at a MOI of 0.01. At 20 h post-infection, the cell cultures were harvested for TCID_50_ assay. Data are presented as means ± SDs from three independent experiments, * *p* < 0.05.

**Figure 5 viruses-13-01866-f005:**
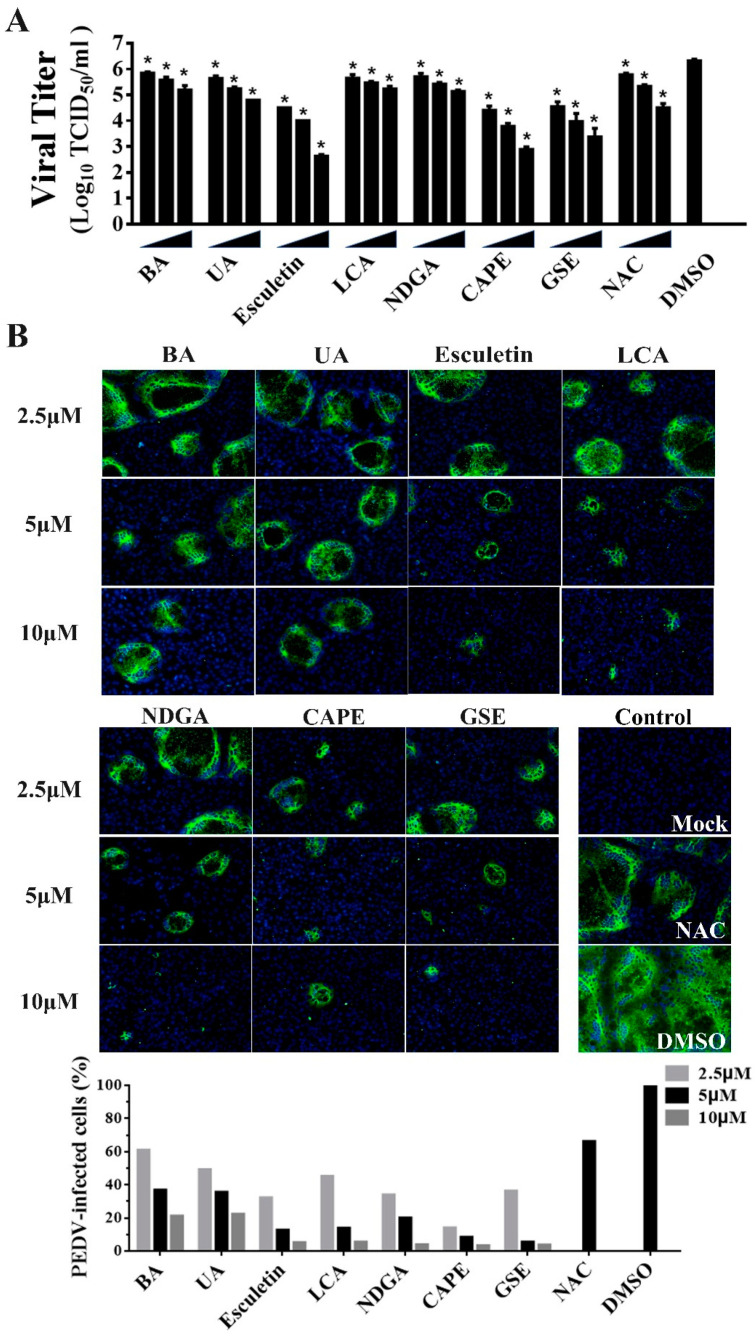
The inhibitory effects of seven kinds of antioxidants on PEDV infection on Vero cells. Vero cells were incubated with seven antioxidants at various concentrations (2.5, 5, and 10 μM) for 1 h, and then infected with wild-type PEDV at a MOI of 0.01. (**A**) At 20 h post-infection, the titers of infected cell cultures were determined by TCID_50_. NAC (50, 100, and 200 μM) and DMSO were used as the positive and negative controls. Data are presented as means ±SDs from three independent experiments. (**B**) PEDV S protein expression (green) in infected cells was analyzed by IFA using anti-PEDV S mAb 4B2. Nuclei (blue) were stained with DAPI. The percentage of PEDV-infected cells was counted based on the IFA results (bottom). NAC (50 μM) and DMSO were used as the positive and negative controls, * *p* < 0.05.

**Figure 6 viruses-13-01866-f006:**
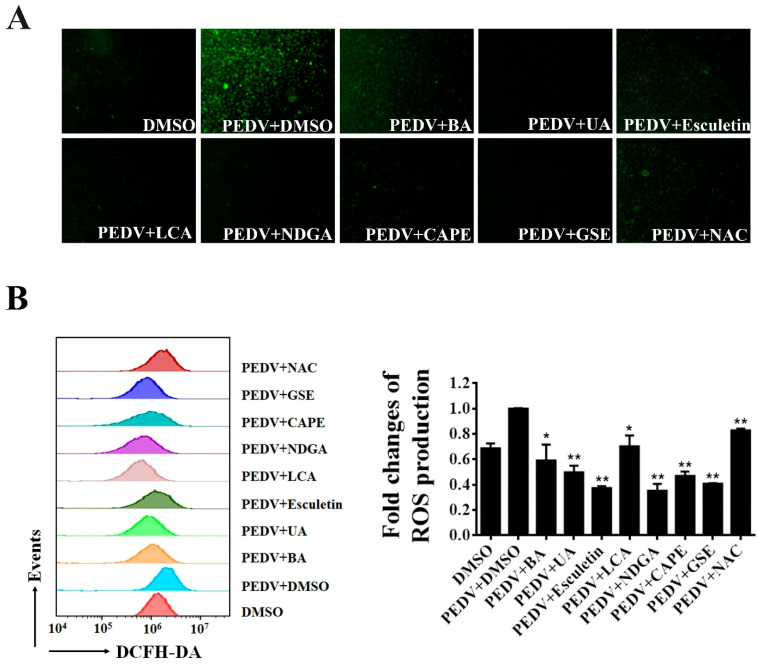
The inhibition of PEDV-induced ROS by identified antioxidants. Vero cells were treated with each antioxidant at the concentration of 10 μM, and then infected with wild-type PEDV at a MOI of 0.01. NAC (50 μM) and DMSO were used as the positive and negative controls. At 16 h post-infection, the cells were stained with DCFH-DA and intracellular ROS generation was examined by a fluorescence microscope (**A**) or flow cytometry (**B**). The relative ROS levels were calculated by the DCFH-DA fluorescence intensity in PEDV-infected cells. Mean ± SDs from three independent experiments are displayed. * *p* < 0.05 and ** *p* < 0.01 vs. PEDV-infected cells treated with DMSO (unpaired Student’s *t*-test).

**Table 1 viruses-13-01866-t001:** The IC_50_s, CC_50_s and SIs of 25 hit compounds.

Compound Name	IC_50_ (μM)	CC_50_ (μM)	SI
Emodin	2.1	>100	>50
gossypol-Acetic acid	2.9	>100	>30
Gynostemma Extract	2.7	>100	>30
Oridonin	3.0	35	10.16
Licochalcone A	4.0	>100	>25
Amphotericin B	2.91	>100	>34.42
Demethylzeylasteral	2.37	38.6	16.27
Tubeimoside I	4.21	74.8	17.76
Harmine hydrochloride	1.33	>100	>75.04
Betulonic acid	<1.25	61.9	>49.52
Ursonic acid	2.13	41	19.23
3′-Hydroxypterostilbene	4.29	>100	>23.29
Tannic acid	4.37	>100	>22.89
(E)-Cardamonin	2.15	>100	>46.44
Harmine	1.96	>100	>51.10
Esculetin	5.97	>100	>16.75
Lithocholic acid	2.37	>100	>42.12
Nordihydroguaiaretic acid	5.00	>100	>19.99
Efonidipine	5.58	>100	>17.93
Tabersonine hydrochloride	4.30	82.6	19.23
Protoporphyrin IX	<1.25	>100	>80
Proanthocyanidins	2.19	>100	>45.71
Caffeic Acid Phenethyl Ester	1.74	>100	>57.63
Grape seed Extract	2.42	>100	>41.37
7-Ethylcamptothecin	<1.25	>100	>80

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
