# Peer review of "Construction of a Recombinant Porcine Epidemic Diarrhea Virus Encoding Nanoluciferase for High-Throughput Screening of Natural Antiviral Products"

_viruses, 2021, doi:10.3390/v13091866_

Round 1
Reviewer 1 Report
The manuscript Viruses-1367576 by Li et al described construction of a PEDV infectious cDNA clone, manipulation of the infectious cDNA clone to replace ORF3 with nanoluciferase (NLuc) gene, and the application of the rPEDV-NLuc virus to screen antiviral products against PEDV. The manuscript was well written and easy to understand. This reviewer only has some minor comments.
- The authors chose to construct an infectious cDNA clone of the PEDV YN150 strain which is a highly cell culture-adapted virus strain (many passages in cell culture). Not sure if the phenotypes of the YN150 strain have dramatically changed when compared to the parental YN1 strain. For screening antiviral products against PEDV, will it make more sense to use a low passage PEDV strain? Please explain and justify the design.
- Lines 222-223: The authors indicated that the rPEDV virus has five genetic markers as shown in Fig 1C. Please clarify whether these five genetic markers were introduced on purpose or accidentally introduced during the cDNA clone construction. Also, please indicate the genes corresponding to these five genetic markers.
- It is not uncommon that EGFP gene can be successfully inserted into a virus genome but EGFP is not stable upon subsequent passages in cell cultures. This reviewer ever inserted EGFP gene into one virus infectious cDNA clone (not PEDV). However, upon serial passages in cell culture, some viruses gradually lost EGFP gene and converted to the parental virus although it was still possible to detect EGFP expression during serial passages of the virus. For example, at the passage 8, EGFP expression was still present; however, if you pick 20 virus plaques for individual examination, most of them have lost EGFP and only a few virus plaques still contained EGFP. This reviewer has never inserted NLuc to virus and is not sure about its stability. On Lines 236-237, the authors stated that “Notably, we observed that rPEDV-NLuc retained the inserted NLuc gene up to five passages in Vero cells (data not shown)”. This reviewer is curious what the authors have done to evaluate the NLuc stability in the rPEDV-NLuc virus. Did they notice the changes of NLuc expression levels in viruses at different passages? Did they pick multiple virus plaques at different passages to assess if all virus plaques still contain NLuc gene? Stability of NLuc in the rPEDV-NLuc virus is important to determine if this recombinant virus can be a reliable tool for screening antiviral products against PEDV.
Reviewer 2 Report
Manuscript review: viruses-1367576 “Construction of a recombinant porcine epidemic diarrhea virus encoding nanoluciferase for high-throughput screening on natural antiviral products.” Corresponding author: Rui Luo.
The authors develop a PEDV reporter virus to for HTS screening for antiviral compounds. This reporter virus also has the potential to be a tool to unravel the mechanisms of PEDV pathogenesis. Overall the paper contributes to the field and is well written, but there are some issues to be addressed prior to publication. I am left with questions that I believe can be easily clarified in the text.
Comments:
- Figure 1B: Please clarify in the legend how many hours post-infection these images were taken. For both Figures 1B and 2B please indicate the magnification.
- Figure 3: 1) why is the plaque morphology between rPEDV and rPEDV-Nluc so different in this figure, when they look exactly the same in figure 2E? 2) I think Fig 3B might be mislabeled, because the highest luminescence (red) is labeled MOI 0.01 and the lowest (blue) is labeled MOI 0.04. 3) I like Fig D showing correlation between luminescence and MOI. Could the authors include luminescence vs titer? In my experience with NLuc reporter viruses, the luciferase virus is magnitudes more sensitive compared to its rg-control, and it would be interesting to see if that holds true for PEDV.
- Figure 5B: It would strengthen the data to quantify % of infected cells from these images. Also, please include magnification in the legend.
- Figure 6: In 6B the shift in histogram seems very minor between mock and NAC. Why are the histograms of the mock (DMSO? Are they uninfected?) treated cells so intense? That seems high – how was the limit (dotted line) determined? Please clarify this figure.
- Figures 1B and 2B: The images look like grey squares and I am not able to clearly see the CPE. If I turn the contrast all the way up on my computer I can see the cells faintly. I also can’t see the cells when I print the manuscript. I am not sure if this is a problem with my devices but I can’t see the CPE.

Reviewer 3 Report
This is an outstanding manuscript on the reverse-genetics experiments on porcine epidemic diarrhea virus (PEDV). The authors created an infectious clone for further testing of inhibitory effect of naturally collected plant extracts. The data are clearly presented and support the primary hypothesis. The are other papers focusing inhibitory effect of plant extracts against PEDV but this one is especially clear and well-written.
I suggest to accept this paper in its present form.
Author Response
We really appreciate the review positive comments and encouragement.